# Low-Protein Formulas with Alpha-Lactalbumin-Enriched or Glycomacropeptide-Reduced Whey: Effects on Growth, Nutrient Intake and Protein Metabolism during Early Infancy: A Randomized, Double-Blinded Controlled Trial

**DOI:** 10.3390/nu15041010

**Published:** 2023-02-17

**Authors:** Ulrika Tinghäll Nilsson, Olle Hernell, Bo Lönnerdal, Merete Lindberg Hartvigsen, Lotte Neergaard Jacobsen, Anne Staudt Kvistgaard, Pia Karlsland Åkeson

**Affiliations:** 1Department of Clinical Sciences, Pediatrics, Lund University, 221 00 Lund, Sweden; 2Department of Clinical Sciences, Pediatrics, Umeå University, 901 87 Umeå, Sweden; 3Department of Nutrition, University of California, Davis, CA 95616, USA; 4Arla Foods Ingredients Group P/S, 8260 Viby J, Denmark

**Keywords:** alpha-lactalbumin, amino acids, CGMP, energetic efficiency, infant growth, infant formula, low protein, obesity, protein metabolism, protein quality

## Abstract

Protein intake is higher in formula-fed than in breast-fed infants during infancy, which may lead to an increased risk of being overweight. Applying alpha-lactalbumin (α-lac)-enriched whey or casein glycomacropeptide (CGMP)-reduced whey to infant formula may enable further reduction of formula protein by improving the amino acid profile. Growth, nutrient intake, and protein metabolites were evaluated in a randomized, prospective, double-blinded intervention trial where term infants received standard formula (SF:2.2 g protein/100 kcal; *n* = 83) or low-protein formulas with α-lac-enriched whey (α-lac-EW;1.75 g protein/100 kcal; *n* = 82) or CGMP-reduced whey (CGMP-RW;1.76 g protein/100 kcal; *n* = 80) from 2 to 6 months. Breast-fed infants (BF; *n* = 83) served as reference. Except between 4 and 6 months, when weight gain did not differ between α-lac-EW and BF (*p* = 0.16), weight gain was higher in all formula groups compared to BF. Blood urea nitrogen did not differ between low-protein formula groups and BF during intervention, but was lower than in SF. Essential amino acids were similar or higher in α-lac-EW and CGMP-RW compared to BF. Conclusion: Low-protein formulas enriched with α-lac-enriched or CGMP-reduced whey supports adequate growth, with more similar weight gain in α-lac-enriched formula group and BF, and with metabolic profiles closer to that of BF infants.

## 1. Introduction

Early nutrition influences growth and health in infancy and can impact long-term health and well-being. Breast milk, containing all nutrients in optimal proportions, is the best source of nutrition for the rapidly growing infant [1,2]. Compared to formula-fed (FF) infants, breast-fed (BF) infants have a lower risk of becoming overweight and obese during childhood and adolescence, as well as of associated cardiometabolic diseases, including diabetes type 2 later in life [3,4]. Furthermore, being BF reduces the risk of infections during infancy [5], and of developing asthma later in life [6]. Despite WHO recommendations, less than 50% of infants globally are exclusively BF during their first six months of life [5], and are therefore dependent on infant formula for their nutrition from early infancy. 

According to the current EU regulation, protein concentration in commercial infant formulas should range from 1.8 g/100 kcal to 2.5 g/100 kcal [7], as compared to 1.5 g/100 kcal in mature breast milk [8]. A higher protein content in infant formulas than in breast milk has been deemed necessary due to the potential risk of essential amino acid inadequacy [9]. However, higher protein concentration is likely a contributing factor to the higher weight gain in FF compared to BF infants. According to the early protein hypothesis [10,11], protein overload leads to higher concentrations of branched-chain amino acids (BCAA), resulting in increased secretion of insulin and insulin growth factor 1 (IGF-1), which may result in accelerated weight gain and deposition of fat, and thus in early metabolic programming of adiposity. Protein content in infant formulas has been reduced over the years based on research by us and other research groups, and international regulations have been revised [12,13,14,15,16,17,18,19,20,21]. However, protein concentration in formula is still considerably higher than in breast milk. 

Infant formula composition has continuously changed over the years to resemble that of breast milk, the gold standard. Hence, the whey to casein ratio in infant formulas has been adjusted from 20/80 in bovine milk to 60/40 in infant formulas. Alpha-lactalbumin, the major whey protein in breast milk accounting for approximately 25% of the total protein content (2.5–3.0 g/L), or about 36% of the whey protein, is still low in standard infant formulas due to its lower concentration in bovine milk, i.e., 3.5% of total protein, or 17% of the whey protein [22,23,24]. However, since α-lactalbumin in human and bovine milk exhibits a similar and favorable amino acid composition with abundant essential amino acids, especially tryptophan and cysteine, increasing its concentration in infant formula could enable further reduction of protein concentration with a protein composition and amino acid profile more similar to that of breast milk [22]. Previous clinical studies have shown that infant formula enriched with α-lactalbumin supports age-appropriate growth [25,26,27,28,29,30], with increased energetic efficiency [25] and improved gastrointestinal tolerance [26,31].

An alternative way to lower the total formula protein concentration is to use a whey fraction reduced in casein glycomacropeptide (CGMP). CGMP is a cleavage-product of bovine ĸ-casein, which contributes to an unfavorable amino acid profile with lack of essential aromatic amino acids such as tryptophan, phenylalanine, tyrosine and also cysteine, but is instead rich in threonine, proline, glutamine and serine [32]. 

However, addition of CGMP-reduced whey could thus allow a reduction of total protein content, still providing adequate concentrations of essential amino acids. 

The primary objective of this study is to evaluate the effect on growth of infants between two and six months of age fed protein-reduced infant formula, either enriched in α-lactalbumin or reduced in CGMP. Our secondary objectives are to evaluate possible effects on nutrient intake and energetic efficiency, as well as on markers of protein metabolism, hematologic parameters, gastrointestinal tolerance, crying time, and time to fall asleep. 

## 2. Materials and Methods

### 2.1. Study Design and Population

The ALFoNS study is a prospective, double-blinded, randomized controlled, intervention trial (RCT) with parallel design comparing three groups of infants fed either standard infant formula (SF) with total protein content of 2.20 g/100 kcal (10% α-lactalbumin of protein), or reduced protein experimental infant formulas with total protein content of 1.75 g/100 kcal with 27% α-lactalbumin of protein (α-lac-EW), or 1.76 g/100 kcal with 14% α-lactalbumin of protein and reduced level of CGMP (CGMP-RW). Exclusively breast-fed infants served as a reference group (BF). 

The study was conducted at two study sites: Malmö/Lund from December 2014, and extended to Umeå from June 2018 due to lower recruitment rate than expected. The intervention was completed in March 2020. Infants were recruited by invitation letters sent to all families with a four-week-old infant in Malmö and Lund and surrounding area (~8500 deliveries/year) and in Umeå (~1800 deliveries/year). Interested families were informed about the study by telephone by the study nurses. On meeting the inclusion criteria, the parents were invited to the pediatric research unit at the respective study site. Infants were included from five to eight weeks of age in order not to interfere with the establishment of breastfeeding. 

Inclusion criteria were healthy term infants (37–42 weeks gestational age) appropriate for gestational age, with a birth weight of 2500–4500 g, i.e., within ±2 SD of the national growth chart. At least one parent should be able to read and communicate in Swedish. At inclusion, infants in the formula groups should be exclusively FF, and infants in the BF group exclusively BF, with the mother’s intention to continue either exclusive formula feeding or breastfeeding until six months of age. However, tiny taste portions of complementary food were allowed from four months of age, in accordance with the Swedish National Food Agency recommendations [33]. Exclusion criteria were neonatal problems, malformations, disabilities, or any other diseases that could interfere with normal nutrition or growth, as well as infants with feeding problems or infantile colic. In addition, infants delivered by Caesarean section or having received antibiotic treatment prior to inclusion were excluded. 

### 2.2. Primary and Secondary Outcomes

Primary outcome was growth, and secondary outcomes were nutrient intake and energetic efficiency, as well as markers of protein metabolism, hematologic parameters, gastrointestinal tolerance, crying time, and time to fall asleep.

### 2.3. Sample Size and Statistical Power

Sample size was calculated to include 80 infants (40 boys and 40 girls) in each group, allowing a loss to follow-up of 20%, to be able to detect a difference in weight of 400 g (0.5 SD) between the study groups at six months of age, with 80% power at a significance level of *p* < 0.05. Due to a possible risk of a higher loss to follow-up than 20% at six months, we decided to include an additional eight infants in order to maintain enough power. Thus, a total of 328 infants, 245 FF (122 girls, 123 boys) and 83 BF (43 girls, 40 boys) were finally included. The intention to treat (ITT) population consisted of all infants who stayed in the study until six months of age, and the per protocol population (PP) were those who fulfilled all protocol requirements. A subpopulation of 200 infants (50 infants (25 boys/25 girls) from each FF group, and 50 BF infants (25 boys/25 girls)) were randomly selected, if meeting the criteria of having completed intervention per protocol with complete dietary records as well as blood, urine and fecal samples at baseline, four and six months of age, for analyses of blood urea nitrogen and amino acids.

### 2.4. Randomization and Blinding

Formula-fed infants were stratified by gender and assigned with random blocks of 6 or 12 using a computerized allocation sequence to receive either SF, α-lac-EW or CGMP-RW from inclusion to six months of age, i.e., during the intervention period. FF and BF infants were consecutively included in a 3:1 pattern with equal numbers of boys and girls. The only exception was at the beginning of the study when every infant meeting the inclusion criteria, regardless of the type of feeding, was included to get the study started. After the inclusion of 10 BF boys and 10 BF girls, FF infants were prioritized to achieve the inclusion rate of 3:1 of FF and BF infants. Blinding of infant formula was done at the formula production site (Laiterie de Montaigu, Le Planty, France). There was no difference in taste or smell among the three infant formulas, which were packed in identical cans, and only differed by a specific code. The double-blinded allocation was retained until all infants had completed their six months visit and all laboratory and statistical analyses up to six months of age had been performed. 

### 2.5. Study Infant Formulas

Study formulas were produced at Laiterie de Montaigu; and Lacprodan^®^ ALPHA-10, Lacprodan^®^ DI-8090 and Lacprodan^®^ DI-8095 were provided by Arla Foods Ingredients Group P/S, Denmark. Study formulas were provided free of charge to the families by the study nurses. Energy and protein content of the study formulas are presented in Table 1. Detailed nutrient content is presented in Appendix A, and amino acid content in Appendix A. Detailed instructions on how to prepare the infant formulas were given to the parents both orally and in writing. 

### 2.6. Study Visits, Growth Parameters, Dietary and Symptom Records

Infants visited the study sites at baseline (five to eight weeks), and at three, four, five and six months of age. At baseline, perinatal and background data on infants and parents were collected. Growth parameters were measured at each visit. Weight was recorded with an accuracy of 5 g (Malmö/Lund: UWE AIN 3 or TANITA BD-815MA, Umeå: SECA 757), and recumbent length (all sites: SECA 416) and head circumference (all sites: non-stretchable measuring tape, SECA 212) measured with the accuracy of 1 mm. Growth velocity (gains in weight, length, and head circumference) during the intervention period was calculated by dividing the increase in anthropometric measurements by the exact number of days between visits at baseline, four and six months of age for each infant. Weight gain is presented as g/d, and gain in length and head circumference as cm/month. Age-adjusted z-scores were calculated for growth parameters using the WHO growth standard [34,35].

Parents were asked to complete a detailed three-day dietary diary before each monthly visit. Infant formula intake was reported in ml/meal and complementary food intake in volume (ml or dl), weight (g) or as number of tea- or tablespoons (5 or 15 mL). Infants were fed on demand. Macronutrient and energy intake from complementary food was calculated by a dietician using the Dietist Net Pro^®^ (Kost och Näringsdata AB, Bromma, Sweden) database, including nutritional data from national and international food agencies and baby food manufacturers. Average daily energy and protein intake from infant formula feeding were calculated at three, four, five and six months of age. Energetic efficiency of the infant formula was calculated as the ratio of weight gain and length gain from two to four months, four to six months and from two to six months, to the average total energy or protein intake during the same period, and expressed as weight gain per 100 kcal or per g protein, and as length gain per 100 kcal or per g protein. 

Stool frequency and consistency (hard, soft, firm, or watery) were registered daily as well as the occurrence of vomiting (not regurgitation), stomach pain, flatulence, any illness, doctor´s visit or medication. Time to fall asleep (<5 min, 5 to <15 min, 15 to <30 min, 30 to <60 min and >60 min), and daily crying time (<30 min, 30 to <60 min, 60 to <120 min, 120 to <180 min and ≥180 min) were recorded as predefined time intervals and registered two days every week. The relation of crying in connection with feeding was also registered.

### 2.7. Blood Sampling and Biochemical Analyses

Venous blood samples were collected at baseline and at four and six months of age at least 2 h postprandially. Local anaesthetic topical cream was used prior to sampling. Hemoglobin (Hb) was analyzed within a few hours at the University and Regional Laboratories of Skåne or at Norrland´s University Hospital on a Sysmex XN-10 instrument (Sysmex Corporation, Chuo-ku, Kobe, Japan). Blood was centrifuged at 1300× *g* for 10 min. Serum was pipetted into separate tubes depending on the type of analysis and was then stored at −80 °C at the Biobank in Lund or Umeå, respectively, until analysis. When the last child had completed their intervention period at six months of age, samples were transported frozen on dry ice to the respective laboratory for analysis. 

Serum iron was analyzed on a Cobas 701 instrument (Roche Diagnostics, Basel, Switzerland), serum C-peptide, insulin, and ferritin on a Cobas 601 instrument (Roche Diagnostics), and serum IGF-1 by the IDS-iSYS assay (Immunodiagnostic System Ltd., Boldon, Tyne & Wear, England). All analyses were carried out at the University and Regional Laboratories of Skåne. Anemia was defined as Hb < 90 g/L at baseline and <105 g/L at four and six months of age. Iron depletion was defined as serum ferritin < 40 µg/L at baseline, <20 µg/L at four months, and <12 µg/L at six months [36].

Serum leptin and leptin receptor were analyzed by ELISA (Human Leptin ELISA kit, EMD Millipore; Merck KGaA, Germany and Human Leptin R Quantikine^®^ ELISA, R&D Systems Inc., Minneapolis, MN, USA) at the Pediatric Research Laboratory at Umeå University. 

Samples from the subpopulation of 200 infants were analyzed for blood urea nitrogen (BUN) using a urea nitrogen colorimetric detection kit (Life Technologies Corporation, Fredrick, MD, USA) according to the manufacturer´s instructions at the University of California, Davis (UCD). In the same subgroup, serum amino acids were analyzed by ion exchange column chromatography (IEC). Samples were acidified to 2% sulfosalicylic acid (SSA) final concentration and then diluted with 100 nmol/mL AE-Cys Li diluent prior to the 50 ul injection. Free amino acids were separated using ion-exchange chromatography with post-column ninhydrin reaction. The IEC instrument (Hitachi 8900), buffers and column were from Hitachi Tokyo, Japan. Calibration was performed using amino acid standards (Sigma-Aldrich, St. Louis, MO, USA). The included reference standard (AE-Cys) was used to correct for any variances in injection volume due to the autosampler [37].

### 2.8. Ethical Considerations

This study was conducted in accordance with the Declaration of Helsinki and approved by the Regional Ethical Review board in Lund. Detailed oral and written information about the study was given and written informed consent was obtained from parents/legal caregivers before inclusion. The study was registered at ClinicalTrials.gov (accessed on 29 September 2021) (NCT02410057).

### 2.9. Statistical Analysis

Statistical analyses were performed using SPSS 25.0 (Released 2017. IBM statistics for Windows, Version 25.0. Armonk, NY, USA: IBM Corp). Results are presented as mean ± SD or as median and interquartile range (IQR, 25th and 75th percentile). For group comparisons of normally distributed continuous data, one way-ANOVA, with post hoc Bonferroni test was used. Separate one-way ANOVA for the three FF groups were also performed. In tables and figures, significant differences between groups are marked with a superscript. 

Growth parameters, as well as serum IGF-1, insulin, C-peptide, leptin and leptin receptor were also compared with analysis of covariance (one-way ANCOVA with post hoc Bonferroni test) adjusted for maternal weight gain, smoking during pregnancy, gestational diabetes, and maternal and paternal BMI. For group comparison of variables with skewed distribution, Kruskal–Wallis test with post hoc Bonferroni was used. Chi-square test or Fisher´s exact test were used for categorical data. Significance level was set at *p* < 0.05. All analyses were performed as intention-to-treat (ITT). Outcome was also analyzed in the per protocol (PP) population and reported for weight data when the result was significantly different from the ITT population.

## 3. Results

### 3.1. Study Populations

A total of 328 infants were recruited to the study groups (Figure 1). During the intervention period, 33 FF infants (13%) dropped out of the study, of whom one infant was excluded due to a congenital genetic disease that affected growth. There were no differences in drop-out rates among the formula groups due to gastrointestinal adverse events or cow´s milk protein allergy. In the BF group, ten infants were lost to follow up and six needed to supplement with formula but stayed in the study as part of the ITT population. 

### 3.2. Background and Baseline Characteristics

Background and baseline characteristics of infants and parents are presented in Table 2. Higher education was more common among mothers of BF than of FF infants. BMI at enrollment was lower in mothers of BF than of FF infants. Mothers of BF infants also had lower pre- and post-gestational weight (data not presented). Use of probiotics was lower in the BF group. A few infants had been hospitalized one to two days prior to enrollment for short duration phototherapy due to mild neonatal icterus or upper airway infections, but none had been treated with antibiotics (data not presented). About one third of the parents were diagnosed with allergy or well-controlled asthma, and a few mothers with well-regulated hypothyroidism (data not present).

### 3.3. Nutrient Intake

Information on formula intake was available for 210 infants (99%) at five months and for 203 infants (96%) at six months in the ITT population (Table 3). Intake of study formula (ml/kg/day) was significantly higher in SF than in α-lac-EW and CGMP-RW infants at three, four and five months, but not at six months, whereas the number of daily meals of formula was the same except at six months where infants in the α-lac-EW group had more meals than infants in the CGMP-RW group. 

Energy intake from formula was higher in the SF than in the α-lac-EW and CGMP-RW groups at three and four months, and close to significantly higher (*p* = 0.053) at five months than in the CGMP-RW group. Protein intake from formula was higher in the SF group than in both α-lac-EW and CGMP-RW groups during the intervention period (Table 3). 

At five months, complementary food intake was low (≤2 tablespoons per day) in 85% and 84% of FF and BF infants, respectively, whereas at six months the corresponding numbers were 64% and 69%. Energy and protein intake from complementary foods, as well as total energy intake, were similar among the formula groups at five and six months of age.

### 3.4. Growth

Data on growth among the ITT population is presented in Table 4. At baseline, weight, length, and head circumference did not differ between the study groups. At four to six months, no differences were found in weight or length among the FF groups. All FF groups had higher growth rates than the BF group, except between four and six months where weight gain was similar in α-lac-EW and BF infants. Between two and four months, length gain was higher in α-lac-EW than in BF infants (2.93 ± 0.48 cm vs. 2.71 ± 0.47 cm, *p* = 0.042). In the PP population, absolute weight between two and six months and weight gain between four and six months were similar in α-lac-EW and BF infants (Table 5). 

*Z*-scores for growth, presented in Figure 2, were within the WHO reference ranges for weight, length and head circumference during the intervention period in all study groups. At baseline, there were no differences between the groups in weight-for-age *z*-score (WAZ), length-for-age *z*-score (LAZ), weight-for-length *z*-score (WLZ), head circumference-for-age *z*-score (HCAZ) or BMI-for-age *z*-score (BMIAZ). At 4 and 6 months, WAZ, and at 4 months WLZ were similar in all FF groups, but significantly higher than in the BF group. At 6 months, WLZ was similar in α-lac-EW and BF infants and lower than in CGMP-RW and SF. LAZ and HCAZ were similar in all study groups at 4 and 6 months of age. BMIAZ was similar in all FF groups and higher than in BF infants at four and six months, except at six months where similar in α-lac-EW and BF infants.

When adjusted for weight gain during pregnancy, gestational diabetes, maternal smoking during pregnancy, maternal BMI, paternal BMI and WLZ (2 mo), WLZ at 4 and 6 months was similar in SF, α-lac-EW and BF groups, but higher in CGMP-RW than in BF infants.

### 3.5. Energetic Efficiency

Weight gain (g) as well as length gain (mm) per 100 kcal of consumed infant formula was similar in all formula groups between two and four months, four and six months and between two and six months (Table 6), but length gain per 100 kcal tended to be higher in α-lac-EW than in SF infants at two to four months and two to three months (0.18 ± 0.05 vs. 0.16 ± 0.05, *p* = 0.07, PP population). Weight gain (g) and length gain (mm) per g protein intake, were higher in α-lac-EW and CGMP-RW, than in SF infants at the same time points (Table 6).

### 3.6. Biochemical Analyses

At baseline, serum BUN was higher in SF and α-lac-EW than in BF infants (Table 7). At four and six months, serum BUN was similar in α-lac-EW and CGMP-RW compared to BF, but lower than in SF infants. 

Serum concentrations of amino acids are presented in Figure 3 and Appendix A. At four months, serum isoleucine and valine were similar in CGMP-RW and BF. Serum valine was higher at four and six months in the SF group compared to all other groups, and higher in α-lac-EW than in CGMP-RW. At four and six months, serum isoleucine was higher in SF than in CGMP-RW and BF infants, and higher in α-lac-EW than in CGMP-RW infants. Serum leucine at four and six months was similar in α-lac-EW and BF, but lower than in SF and CGMP-RW infants. Serum tryptophan (Trp) was similar at baseline, four and six months in the formula and BF groups, except at six months when it was higher in CGMP-RW than in BF infants. Serum histidine (His) was higher at baseline but lower at six months in all FF compared to BF infants. At four months, serum threonine (Thr) was similar in CGMP-RW and BF, and at six months similar in CGMP-RW and BF as well as in SF and CGMP-RW, but higher in α-lac-EW than in CGMP-RW and SF infants at both four and six months. 

Mean total branched chain amino acids (BCAA—sum of isoleucine (Ile), leucine (Leu) and valine (Val)) was higher at four and six months in FF than in BF infants, but lower in α-lac-EW and CGMP-RW than in SF infants (Figure 4).

Serum insulin and C-peptide were similar among the FF groups, and higher than in BF infants at baseline (except in α-lac-EW where insulin was similar to BF) and during the intervention period (Table 7). Serum IGF-1 was similar at baseline in all groups (except in α-lac-EW where S-IGF-1 was higher than in BF), but at four and six months was higher in all FF groups compared to BF infants. Weight gain per day was positively correlated with the difference in S-IGF1 concentration between two and six months in all FF infants (r = 0.32, *p* < 0.001), and also when analyzing the FF groups separately; SF (r = 0.26, *p* = 0.004), α-lac-EW (r = 0.31, *p* = 0.014), CGMP-RW (r = 0.37, *p* = 0.004), but no correlation was found among BF infants (r = −0.076, *p* = 0.56).

Serum leptin at baseline, four and six months, was similar in all groups and no difference was found between boys and girls (Table 7). Soluble leptin receptor (SLR) was higher at baseline in BF compared to FF, but similar at four and six months in all groups. Free leptin index (leptin/SLR) was lower in SF than in CGMP-RW at four months when data were adjusted for gestational diabetes mellitus, weight gain during pregnancy, smoking, and maternal and paternal BMI (Table 7). 

Hemoglobin (Hb) was similar in all groups at four and six months (Table 7). There was no difference in serum iron concentration among the FF groups at any age. Higher serum iron was found in α-lac-EW at four months, and in CGMP-RW at six months, compared to BF infants. At baseline, serum ferritin was similar in all FF groups, but higher in CGMP-RW than in SF infants at four months. At baseline, serum Hb was less than 90 g/L in four FF infants, and at four and six months less than 105 g/L in seven infants (six FF (two in each group)), and one BF infant), but none had signs of iron depletion. 

### 3.7. Gastrointestinal Symptoms

All three formulas were well tolerated with regard to gastrointestinal symptoms (Appendix A). Frequency of vomiting was similar among all FF groups, but less common in the BF group. There was no difference in frequency of stomach pain or flatulence among the groups during the intervention. There was no difference in number of or consistency of stools among the FF groups, although BF infants had a higher number of stools and lower frequency of hard and firm stools compared to the FF groups.

Median frequency of days with crying related to feeding was low in general and similar in all groups (Appendix A). Most infants cried less than 30 min per day, and it took less than 15 min for the infant to fall asleep, with no difference among the groups. 

### 3.8. Adverse Events

Adverse events (AE) were reported in 182 FF infants (SF = 64, α-lac-EW = 58 and CGMP-RW = 60) and in 57 BF infants, with no significant differences among the groups, the majority relating to fever or mild airway infections (defined as ≥ 2 of the following symptoms; cough, breathing difficulties, nasal congestion, runny nose or fever). Hospitalization was reported in six infants, three in the CGMP-RW group; RSV bronchiolitis or fever of unknown origin, and three in the SF group; pylorus stenosis, perianal abscess (both underwent surgical intervention), and one apneic episode. Gastrointestinal adverse events (watery or hard stools, stomach pain, vomiting or flatulence), including those diagnosed with CMPA, that resulted in discontinuation of the study or switching to another formula were reported in 26 infants (11 in SF, 4 in α-lac-EW and 11 in CGMP-RW group), but with no significant difference among the FF groups (*p* = 0.12). 

## 4. Discussion

By increasing the proportion of α-lactalbumin-enriched whey or CGMP-reduced whey to infant formula, our study shows that the total protein concentration can be safely lowered to 1.75 g/100 kcal, which is slightly below the minimum regulation of 1.8 g/100 kcal according to the EU directive [7]. Addition of α-lactalbumin-enriched or CGMP-reduced whey improved protein quality, as shown by similar BUN concentrations and similar or higher concentrations of serum essential amino acids, including tryptophan, in infants fed low-protein formulas and in BF infants. In addition, weight gain was similar in infants fed α-lactalbumin-enriched formula compared to BF infants between four and six months. Thus, our study supports two safe alternative ways of improving protein quality when reducing infant formula total protein concentration.

Feeding infants formula with higher protein concentration has been shown to result in higher weight gain than if low-protein formula or breast milk were given [12,14,19]. In the present study, growth was similar in infants fed SF or experimental formulas (EF), and similar or higher than in BF infants. The lack of difference in growth rates between the SF and EF infants may be explained by the lower protein concentration in SF compared to other studies [14,19], as well as the relatively small differences between standard and experimental formulas. Our findings of similar weight gain and Δ WAZ between four and six months in infants fed α-lac-EW compared to BF infants, as well as similar weight at two to six months in these groups in the PP analyses, may indicate similarities in the nutrient and protein composition of EF enriched with α-lactalbumin as compared to breast milk. The higher length gain and tendency to a higher energetic efficiency (length gain per 100 kcal) in the α-lac-EW group compared to the SF group may also have resulted from improved protein quality achieved by the addition of α-lactalbumin. Our findings are consistent with a recent study where infants were fed low protein formula (1.43 g/100 kcal) with an increased proportion of α-lactalbumin (26%) between one and four months of age [27], and another study showing increased energetic efficiency after the addition of α-lactalbumin [25].

Our findings of lower ingested volumes of formula up to five months of age in EF compared to SF groups, resulting in lower energy intakes, despite practically isocaloric formulas, may indicate higher satiety among infants fed formulas with added α-lactalbumin-enriched or CGMP-reduced whey compared to those fed SF, also resulting in further reduced protein intakes. However, we did not find any differences in the appetite regulating hormone leptin. Another possibility is that the higher potential renal solute load (PRSL) of SF resulted in increased thirst in this group, and thus a higher volume intake.

The lower serum BUN in infants fed EF compared to those fed SF, and the similar serum BUN in EF groups compared to BF infants in the present study, are not only the result of a lower protein intake in EF infants but could also indicate improved quality of formula protein, in accordance with other studies [30,31], and exemplified by the lower serum branched chain amino acids (BCAAs; isoleucine, leucine, valine), than in the SF group.

BCAAs, considered to increase insulin secretion and thus growth, are traditionally found to be higher in FF compared to BF infants due to the higher protein concentration in formula. Feeding lower protein formula consequently results in lower serum BCAA concentrations closer to those of BF infants, as shown by us and others [13,15,38,39]. In the present study, total BCAAs at four and six months were found to be lower in the EF groups than in the SF group, but still higher than in BF infants. Leucine has previously been pointed out to be especially critical, since a high level of leucine may result in lower beta-oxidation, increased adiposity and infant weight gain [38]. Our finding of similar serum leucine at four and six months in the α-lac-EW group compared to BF infants is thus noteworthy and is consistent with a previous study [30]. In addition, by adding CGMP-reduced whey to low protein formula, serum isoleucine and valine concentrations were found to be similar to those of BF infants at four months of age. These lower levels of serum BCAA in the EF groups could probably be explained by the different amino acid composition of the EF formulas compared to standard formula.

Tryptophan is one of the limiting amino acids when lowering protein concentration in infant formula, and is necessary in the formation of serotonin and accordingly melatonin, thereby affecting sleep patterns [22,40]. The addition of α-lactalbumin or CGMP-reduced whey to the EFs in the present study resulted in similar or higher serum tryptophan concentration compared to BF infants. Time to fall asleep in the present study was also similar in all groups and no sleep issues were reported, which may partly be related to the similar and adequate serum tryptophan levels in all FF groups.

In contrast to other studies, serum histidine was marginally lower in all FF groups compared to the BF group during the intervention period, despite histidine concentration in all study formulas being higher than the minimum requirements (40 mg/100 kcal). The reason for this is unclear and is not considered a concern.

Serum C-peptide has been reported to increase with increasing protein concentration in infant formula [13,39], probably because of higher levels of insulinogenic amino acids. Despite lower protein concentrations in both the EFs in the present study, C-peptide was similar in all FF groups and remained higher during the intervention period compared to the BF group. In addition, despite decreasing insulin concentrations in all groups between two and six months, serum insulin did not differ among the FF groups and remained higher compared to BF infants throughout the intervention. Similar concentrations of C-peptide and insulin as found in the FF groups, probably resulted from the relatively small differences in formula protein concentrations. The higher serum C-peptide and insulin concentrations in FF compared to BF infants in the present study are consistent with results from previous studies [29,41,42,43,44,45,46], where infants were fed low protein infant formulas, with or without added α-lactalbumin. Our results suggest that infant formula protein concentration can be further lowered.

Effects of protein intake on growth velocity could be mediated by IGF-1 [10,11], and lower serum-IGF-1 has been found in infants fed low protein formula compared to those fed formula with higher protein concentrations [39]. Despite similar serum IGF-1 at baseline in the FF and BF infants in the present study, higher serum IGF-1 was found in all FF groups compared to the BF group during the intervention, with no difference between EF and SF infants, despite the lower protein concentration in the EF formulas. In a previous study, excessive protein concentration in the standard formula group, 2.9 g/100 kcal from inclusion (~14 days of age) to four months and 4.4 g/100 kcal from four to twelve months, resulted in higher serum IGF-1 at six months when compared to their low-protein group [39], with protein concentration similar to standard formula in the present study. Thus, considerable differences in formula protein concentration in other studies could explain the varying results for serum IGF-1. In accordance with our results, no difference in serum IGF-1 was found when feeding lower protein infant formulas with 1.8 g/100 kcal compared to 2.7 g/100 kcal [43], or with 1.7 g/100 kcal compared to 2.1 g/100 kcal [46].

Since serum IGF-1 is the main growth factor during infancy, the positive correlation found between the difference in IGF1 between two and six month and weight gain from two to six months in all FF groups except for the BF group, is an interesting finding. Previous studies have conflicting results regarding IGF-1 levels in BF infants and infant growth [47,48,49,50,51,52]. The difference in hormonal response between BF and FF infants in our study suggests that factors other than the differences in protein intake affect hormonal response and regulation of weight gain during infancy.

Leptin is essential in the complex regulation of food intake and energy metabolism. Previous studies [53,54] have reported higher plasma leptin in BF compared to FF infants during early infancy, but with no differences in anthropometrics, whereas others have found inverse associations between leptin in breast milk and weight-for-length z-scores at four and twelve months [55]. In our study, serum leptin was similar in all groups which is consistent with other reports [29,46]. Soluble leptin receptor is the major leptin-binding protein in circulation, where leptin circulates either in free form or is bound to this receptor. The leptin/soluble leptin receptor index has been proposed to represent free serum leptin, which could thus be related to the infant´s energy intake [56]. To our knowledge, the present study is the first to evaluate the free leptin index (FLI) in FF compared to BF infants after the neonatal period. However, FLI was generally found to be similar in all FF and BF infants, except at four months when FLI was lower in the SF group compared to infants fed CGMP-RW (adjusted model). Further studies are needed to clarify these observations.

Excessive iron intake in healthy infants without iron deficiency may impair infant growth and increase susceptibility to infections [57]. By adding α-lactalbumin, with its capacity of binding minerals including iron, iron absorption may be enhanced, thus allowing the iron content in infant formula to be lowered [9]. However, iron parameters in the present study were mostly similar in all FF groups despite higher or lower concentrations of α-lactalbumin. Even though we found a few infants with hemoglobin concentrations below reference cut-offs, no signs of iron deficiency were observed. In our study, no infant received iron drops in accordance with national Swedish guidelines. According to a recent study, a formula iron concentration of 8 mg/L, as in the present study, can safely be lowered to 2 mg/L without increasing the risk of iron deficiency in healthy term infants [58].

Some studies report improved gastrointestinal tolerance in infants fed α-lactalbumin enriched formula [26,27,31,59]. However, in the present study, all formulas were well tolerated, with no difference among the FF groups, in accordance with a previous study where α-lactalbumin was added [29]. Although high protein infant formula has been associated with more gastrointestinal symptoms, the protein concentration in our SF was low in comparison with other studies, which may also explain why it was well tolerated and with low number of gastrointestinal symptoms reported. Evaluation of gastrointestinal tolerance may also differ between studies. In our study we considered mild regurgitation to be part of normal feeding-related symptoms in infants, only registering vomiting as a GI-symptom if excessive regurgitation or incidental vomiting occurred. Therefore, smaller differences among our groups might not have been registered.

A strength of this study is the randomized controlled trial (RCT) design, with large numbers of subjects and a low dropout rate. In addition, most infants were only introduced to small taste portions of complementary food between four and six months of age, thus limiting its influence on the results at five and six months. Furthermore, to our knowledge this is the first published RCT comparing infant formula with protein concentration slightly below new current regulatory limits (1.8 g/100 kcal) with either a higher concentration of α-lactalbumin-enriched whey or CGMP-reduced whey. A limitation of the study was that infants were not enrolled until five to eight weeks age, in order not to disturb the establishment of breastfeeding, and also that two thirds of the infants in the FF groups had received breast milk to some extent prior to enrollment, which may have affected our results. However, the period of breastfeeding was brief (about two weeks) and did not differ among the FF groups.

In conclusion, feeding infants low protein infant formula enriched with either high concentration of α-lactalbumin-enriched whey or CGMP-reduced whey is safe and well tolerated, providing adequate growth and a metabolic profile closer to that of BF infants. Increasing the concentration of α-lactalbumin further enables a formula protein composition closer to breast milk. Even though growth velocity was similar at some time-points in infants fed low-protein α-lactalbumin-enriched formula and BF infants, overall growth was still slightly higher among infants fed the low-protein infant formulas compared to BF infants as were serum concentrations of BCAAs, IGF-1, insulin, and C-peptide. Thus, protein concentration can probably be further reduced in infant formulas, if protein quality is kept high. Long-term follow-up data on growth and metabolic profiles are desirable to evaluate whether improved infant formula composition will have long-term beneficial effects throughout childhood.

## Figures and Tables

**Figure 1 nutrients-15-01010-f001:**
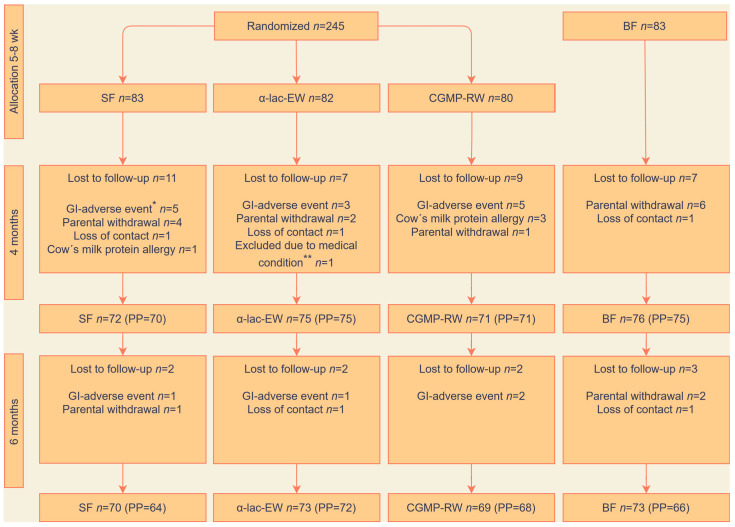
Study flowchart. Randomization, allocation and follow-up in intention-to-treat and per protocol (PP) populations. SF, standard formula; α-lac-EW, experimental formula with α-lactalbumin-enriched whey; CGMP-RW, experimental formula with reduced CGMP whey; BF, breast-fed. Reasons for lost to follow-up: * Gastrointestinal adverse events, such as vomiting, stomach ache, flatulence or constipation. ** Congenital genetic disease affecting growth.

**Figure 2 nutrients-15-01010-f002:**
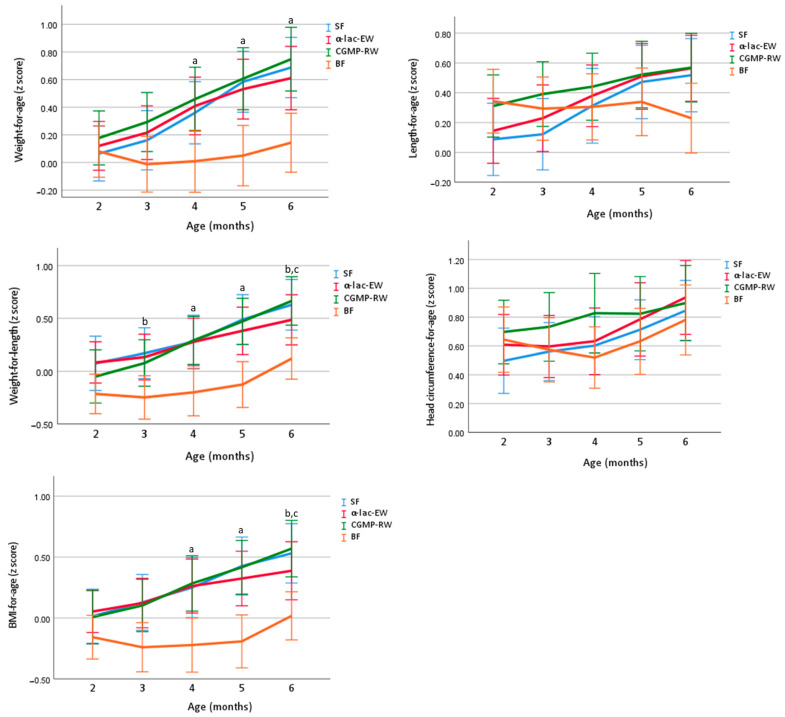
Unadjusted mean (95% CI) of z-score weight-for-age (WAZ), length-for-age (LAZ), weight-for-length (WLZ), HC-for-age (HCAZ) and BMI-for-age (BMIZ) in study groups compared by one-way ANOVA, post hoc Bonferroni. ^a^ all FF groups significantly different vs. BF. ^b^ SF significantly different vs. BF. ^c^ CGMP-RW significantly different vs. BF. SF, standard formula; α-lac-EW, experimental formula with α-lactalbumin-enriched whey; CGMP-RW, experimental formula with reduced CGMP whey; BF, breast-fed.

**Figure 3 nutrients-15-01010-f003:**
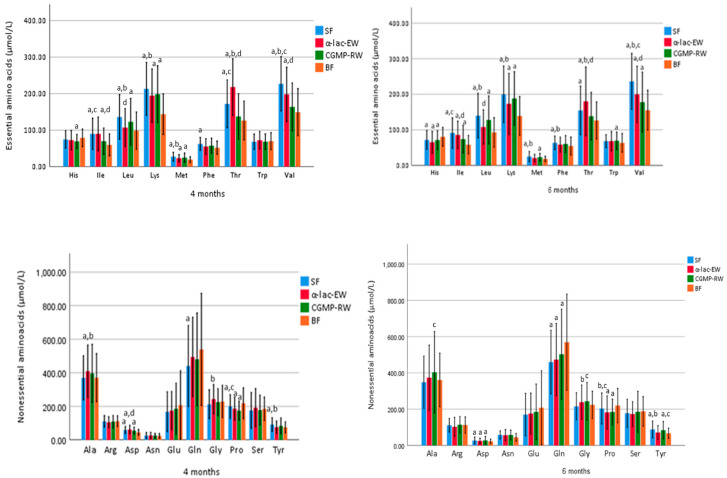
Mean (±SD) plasma concentration of essential and nonessential amino acids in the randomized subgroup (n = 50 in each group) at four and six months of age. Groups compared by one-way ANOVA, post hoc Bonferroni. *p* value < 0.05 are considered statistically significant and marked by superscript letters. ^a^ Significantly different vs. BF. ^b^ SF vs. α-lac-EW. ^c^ SF vs. CGMP-RW. ^d^ α-lac-EW vs. CGMP-RW. SF, standard formula; α-lac-EW, experimental formula with α-lactalbumin enriched-whey; CGMP-RW, experimental formula with reduced CGMP whey; BF, breast-fed.

**Figure 4 nutrients-15-01010-f004:**
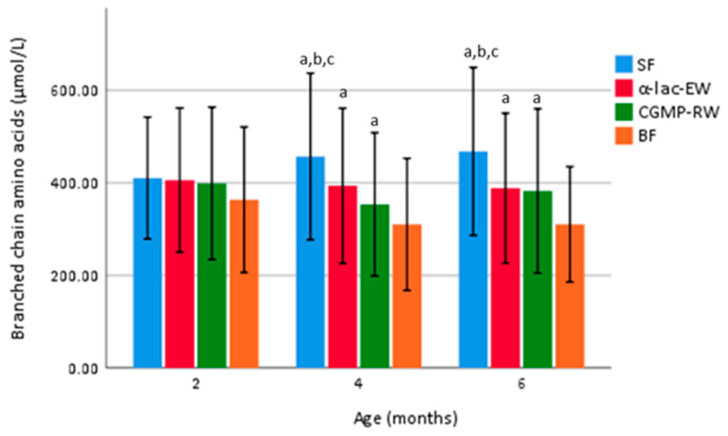
Mean (± SD) total BCCA (isoleucine, leucine, valine) plasma concentration in the randomized subgroup (n = 50 in each group) at four and six months of age. Groups compared by one-way ANOVA, post hoc Bonferroni. *p* value < 0.05 are considered statistically significant and marked by superscript letters. ^a^ Significantly different vs. BF. ^b^ SF vs. α-lac-EW. ^c^ SF vs. CGMP-RW. SF, standard formula; α-lac-EW, experimental formula with α-lactalbumin enriched whey; CGMP-RW, experimental formula with reduced CGMP whey; BF, breast-fed.

**Table 1 nutrients-15-01010-t001:** Energy and protein content in study formulas * and minimum levels according to regulation.

	SF	α-lac-EW	CGMP-RW	Regulation ^1^
Energy (kcal/100 mL)	67.3	68.2	68.0	60
Whey:casein ratio	60:40	70:30	70:30	
Protein (g/100 kcal) Protein (g/100 mL)	2.20 1.48	1.75 1.19	1.76 1.20	1.8
α-lactalbumin (%) ^2^	10	27	14	
CGMP (%) ^2^	9	19	-	

* SF, standard formula; α-lac-EW, experimental formula with α-lactalbumin-enriched whey; CGMP-RW, experimental formula with reduced CGMP whey. ^1^ EU Regulation [7]. ^2^ Percentage of total protein composition. α-lactalbumin was measured in infant formulas, whereas the level of CGMP was estimated.

**Table 2 nutrients-15-01010-t002:** Baseline characteristics of parents and infants fed formula (SF, α-lac-EW or CGMP-RW) * or breast milk (BF), ITT population.

	SF *n* = 83	α-lac-EW *n* = 81	CGMP-RW *n* = 80	*p*-Value 1 All FF **	BF *n* = 83	*p*-Value ^2^ All FF vs. BF
Birth weight (g)	3471 ± 452 ^3^	3527 ± 440	3605 ± 443	0.16	3540 ± 424	0.89
Birth length (cm)	50.2 ± 2.2	50.1 ± 1.9	50.5 ± 1.7	0.42	50.4 ± 1.9	0.51
Birth head circumference (cm)	34.6 ± 1.2	35.0 ± 1.3	35.0 ± 1.2	0.06	35.0 ± 1.5	0.23
Gestational age (wk)	39.5 ± 1.2 ^a^	39.7 ± 1.3 ^b^	40.1 ± 1.2	0.007	40.0 ± 1.1	0.11
Age at inclusion (d)	49.3 ± 5.0	49.4 ± 4.1	49.2 ± 5.8	0.97	50.5 ± 4.5	0.05
Female [*n* (%)]	40 (48)	41 (51)	40 (50)	0.95	43 (52)	0.73
Ever breastfed before inclusion [*n* (%)]	63 (76)	66 (78)	70 (88)	0.17	83 (100)	n/a
Days of breast-feeding before inclusion (*n*)	15.5 ± 14.6	18.0 ± 15.2	17.5 ± 13.9	0.50	50.4 ± 4.6	n/a
Probiotics before inclusion [*n* (%)] ^4^	28 (34)	24 (30)	23 (29)	0.94	13 (16)	0.007
Twins [*n* (%)]	5 (6)	7 (9)	4 (5)	0.63	0 (0)	0.009
Siblings [*n* (%)]	54 (65)	53 (65)	47 (59)	0.61	45 (54)	0.15
Maternal age (y)	31.5 ± 4.8	31.1 ± 4.6	31.1 ± 4.6	0.83	32.6 ± 4.2	0.026
Maternal origin [*n* (%)]						
Nordic	77 (93)	70 (86)	74 (93)	0.29	75 (90)	0.96
European (non-Nordic)	3 (4)	3 (10)	3 (4)	1.0	5 (4)	0.36
Non-European	3 (4)	8 (10)	3 (4)	0.15	3 (4)	0.58
Maternal higher education [*n* (%)] ^5^	45(54)	47 (58)	57 (71)	0.067	66 (80)	0.002
Maternal BMI at enrollment (kg/m^2^)	27.9 ± 5.2	27.9 ± 5.6	26.1 ± 4.2	0.04 ^#^	25.2 ± 3.7	<0.001
Weight gain during pregnancy (kg)	13.1 ± 6.2	13.4 ± 6.6	14.7 ± 6.0	0.20	14.1 ± 5.3	0.59
Gestational diabetes [*n* (%)]	4 (5)	2 (3)	4 (5)	0.66	5 (6)	0.54
Gestational hypertension [*n* (%)]	5 (6)	8 (10)	3 (4)	0.28	5 (6)	0.86
Maternal smoking during pregnancy [*n* (%)]	3 (4)	4 (5)	3 (4)	0.61	0 (0)	0.071
Paternal origin [*n* (%)] ^6^						
Nordic	65 (80) ^a^	68 (86)	73 (94)	0.047	65 (80)	0.17
European (non-Nordic)	6 (7)	3 (4)	2 (3)	0.30	10 (12)	0.015
Non-European	10 (12)	8 (10)	3 (4)	0.15	6 (7)	0.69
Paternal higher education [*n* (%)] ^5^	29 (36)	31 (40) ^b^	46 (59)	0.008	56 (69)	0.052
Paternal BMI (kg/m^2^)	27.0 ± 4.2	27.3 ± 5.8	26.0 ± 4.0	0.24	25.9 ± 4.5	0.18
Paternal smoking during pregnancy [*n* (%)]	14 (17)	11 (14)	5 (7)	0.12	9 (11)	0.71

* SF, standard formula; α-lac-EW, experimental formula with α-lactalbumin-enriched whey; CGMP-RW, experimental formula with reduced CGMP whey.** All FF = infant formula groups combined.^1^ Chi-square test or Fisher´s exact test for categorical data. ANOVA with post hoc Bonferroni for comparison of means. ^2^ Chi-square test or Fisher´s exact test for categorical data. Independent samples t-test for comparison of means. ^3^ Mean ± SD. ^a^ SF vs. CGMP-RW. ^b^ α-lac-EW vs. CGMP-RW. ^#^ ANOVA *p* = 0.04, although post hoc test showed no significant difference between FF groups. ^4^ Probiotics: Sempers magdroppar^®^, wash-out period 7 days before inclusion. ^5^ University or higher professional education. ^6^ Information about baseline characteristics available for 238 fathers.

**Table 3 nutrients-15-01010-t003:** Infant formula, energy and protein intake in infants fed formula (SF, α-lac-EW or CGMP-RW) * during the intervention, ITT population.

		SF		α-lac-EW		CGMP-RW	*p*-Value ^1^
	*n*		*n*		*n*		
Number of meals per day							
3 mo	74	7.0 ± 1.5 ^2^	77	6.9 ± 1.6	74	6.7 ± 1.4	0.55
4 mo	69	6.7 ± 1.6	74	6.8 ± 1.5	71	6.4 ± 1.5	0.31
5 mo	69	6.2 ± 1.5	72	6.6 ± 1.5	69	6.0 ± 1.6	0.11
6 mo	68	5.7 ± 1.6	68	6.2 ± 1.5 ^a^	67	5.4 ± 1.3	0.015
Formula intake per day (mL/kg/d)							
3 mo	74	152 ± 29 ^b,c^	77	140 ± 20	74	141 ± 16	0.003
4 mo	69	144 ± 31 ^b,c^	74	131 ± 18	71	133 ± 18	0.001
5 mo	69	131 ± 23 ^b,c^	72	122 ± 20	69	121 ± 19	0.013
6 mo	68	117 ± 30	68	115 ± 29	67	109 ± 23	0.18
Energy intake per day (kcal/kg/d)							
3 mo	74	102 ± 20 ^b,c^	77	96 ± 13	74	96 ± 11	0.003
4 mo	69	97 ± 21 ^b,c^	74	89 ± 12	71	90 ± 12	0.005
5 mo	69	88 ± 15	72	84 ± 14	69	82 ± 13	0.043 **
6 mo	68	79 ± 20	68	78 ± 20	67	74 ± 15	0.23
Protein intake per day (g/kg/d)							
3 mo	74	2.2 ± 0.4 ^b,c^	77	1.7 ± 0.3	74	1.7 ± 0.2	<0.001
4 mo	69	2.1 ± 0.5 ^b,c^	74	1.5 ± 0.2	71	1.6 ± 0.2	<0.001
5 mo	69	1.9 ± 0.3 ^b,c^	72	1.5 ± 0.2	69	1.5 ± 0.2	<0.001
6 mo	68	1.7 ± 0.4 ^b,c^	68	1.4 ± 0.3	67	1.3 ± 0.3	<0.001

* SF, standard formula; α-lac-EW, experimental formula with α-lactalbumin-enriched whey; CGMP-RW, experimental formula with reduced CGMP whey. ^1^ FF groups compared by one-way ANOVA, post hoc Bonferroni. ^2^ Mean ± SD (all such values). ^a^ α-lac-EW vs. CGMP-RW. ^b^ SF vs. α-lac-EW. ^c^ SF vs. CGMP-RW. ** Post hoc Bonferroni, *p* = 0.053.

**Table 4 nutrients-15-01010-t004:** Anthropometric data and growth velocity in infants fed formula (SF, α-lac-EW or CGMP-RW) * or breast milk (BF) during the intervention, ITT population.

		SF		α-lac-EW		CGMP-RW	*p*-Value ^1 (2)^		BF
	*n*		*n*		*n*			*n*	
Weight (g)									
2 mo	83	5039 ± 5853^3^	81	5092 ± 566	80	5162 ± 604	0.41 (0.47)	83	5093 ± 610
3 mo	75	6209 ± 699	77	6232 ± 712	74	6342 ± 715	0.47 (0.48)	80	6022 ± 723
4 mo	72	7067 ± 780 ^a^	75	7061 ± 834 ^a^	71	7120 ± 847 ^a^	0.89 (0.80)	76	6685 ± 888
5 mo	71	7781 ± 834 ^a^	73	7691 ± 923 ^a^	69	7762 ± 912 ^a^	0.82 (0.80)	74	7263 ± 887
6 mo	70	8323 ± 879 ^a^	73	8228 ± 1025 ^a^	69	8342 ± 1009 ^a^	0.76 (0.67)	73	7771 ± 932
Weight gain (g/d)									
2–4 mo	72	27 ± 6 ^a^	75	27 ± 6 ^a^	71	27 ± 6 ^a^	0.99 (0.98)	76	23 ± 6
4–6 mo	70	20 ± 5 ^a^	73	19 ± 6	69	20 ± 6 ^a^	0.17 (0.15)	73	17 ± 5
2–6 mo	70	24 ± 5 ^a^	73	23 ± 5 ^a^	69	24 ± 5 ^a^	0.59 (0.56)	73	20 ± 5
Length (cm)									
2 mo	83	56.6 ± 2.2	81	56.7 ± 2.0	80	57.1 ± 1.8	0.13 (0.25)	83	57.2 ± 2.0
3 mo	75	60.7 ± 2.3	77	60.9 ± 2.1	74	61.4 ± 2.0	0.14 (0.15)	80	60.9 ± 2.1
4 mo	72	63.7 ± 2.6	75	63.7 ± 2.2	71	63.9 ± 2.0	0.56 (0.85)	76	63.4 ± 2.3
5 mo	71	66.1 ± 2.5	73	66.0 ± 2.4	69	66.1 ± 2.1	0.98 (0.94)	74	65.7 ± 2.3
6 mo	70	67.9 ± 2.5	73	67.9 ± 2.4	69	67.8 ± 2.3	0.99 (0.98)	73	67.1 ± 2.6
Length gain (cm/mo)									
2–4 mo	72	2.9 ± 0.5	75	2.9 ± 0.5 ^a^	71	2.9 ± 0.5	0.78 (0.56)	76	2.7 ± 0.5
4–6 mo	70	2.1 ± 0.6 ^a^	73	2.1 ± 0.5 ^a^	69	2.0 ± 0.4 ^a^	0.84 (0.80)	73	1.8 ± 0.5
2–6 mo	70	2.5 ± 0.4 ^a^	73	2.5 ± 0.3 ^a^	69	2.5 ± 0.3 ^a^	0.54 (0.35)	73	2.3 ± 0.3
HC (cm)^4^									
2 mo	83	38.7 ± 1.3	81	38.7 ± 1.2	80	38.9 ± 1.2	0.53 (0.35)	83	38.9 ± 1.3
3 mo	75	40.6 ± 1.2	77	40.6 ± 1.3	74	40.8 ± 1.4	0.41 (0.49)	80	40.5 ± 1.3
4 mo	72	41.9 ± 1.2	75	41.8 ± 1.4	71	42.1 ± 1.5	0.27 (0.48)	76	41.6 ± 1.3
5 mo	71	42.9 ± 1.2	73	42.9 ± 1.5	69	43.0 ± 1.5	0.95 (0.90)	74	42.8 ± 1.4
6 mo	70	43.9 ± 1.2	73	43.9 ± 1.6	69	43.9 ± 1.6	0.84 (0.71)	73	43.7 ± 1.5
HC gain (cm/mo)									
2–4 mo	72	1.3 ± 0.3	75	1.3 ± 0.3	71	1.3 ± 0.4	0.43 (0.57)	76	1.2 ± 0.3
4–6 mo	70	1.0 ± 0.2	73	1.0 ± 0.5	69	0.9 ± 0.4	0.052 (0.032) ^a^	73	1.0 ± 0.3
2–6 mo	70	1.2 ± 0.2	73	1.2 ± 0.2	69	1.1 ± 0.2	0.57 (0.20)	73	1.1 ± 0.2

* SF, standard formula; α-lac-EW, experimental formula with α-lactalbumin-enriched whey; CGMP-RW, experimental formula with reduced CGMP whey. ^1^ FF groups compared by one-way ANOVA, post hoc Bonferroni. ^(2)^ FF groups compared by one-way ANCOVA, post hoc Bonferroni, adjusted for weight gain during pregnancy, gestational diabetes, maternal smoking during pregnancy, maternal and paternal BMI. ^3^ Mean ± SD (all such values). ^4^ HC = Head circumferences. ^a^ Significantly different vs. BF (*p* < 0.05).

**Table 5 nutrients-15-01010-t005:** Mean weight and weight gain in infants fed formula (SF, α-lac-EW or CGMP-RW) * or breast milk (BF) during the intervention, PP population.

		SF		α-lac-EW		CGMP-RW	*p*-Value ^1 (2)^		BF
	*n*		*n*		*n*			*n*	
Weight (g)									
2 mo	83	5039 ± 585 ^3^	81	5092 ± 566	80	5162 ± 604	0.41 (0.47)	83	5093 ± 610
3 mo	74	6202 ± 700	77	6232 ± 712	73	6336 ± 718	0.48 (0.45)	80	6021 ± 723
4 mo	68	7040 ± 734	75	7061 ± 834	71	7120 ± 847 ^a^	0.83 (0.72)	73	6715 ± 888
5 mo	67	7760 ± 781 ^a^	73	7690 ± 923	68	7756 ± 917 ^a^	0.87 (0.86)	69	7308 ± 888
6 mo	66	8306 ± 823 ^a^	72	8235 ±1031	68	8337 ± 1015 ^a^	0.82 (0.75)	66	7829 ± 940
Weight gain (g/d)									
2–4 mo	68	27 ± 6 ^a^	75	27 ± 6 ^a^	71	27 ± 6 ^a^	0.99 (0.98)	73	23 ± 6
4–6 mo	66	20 ± 5 ^a^	72	19 ± 6	68	20 ± 6 ^a^	0.18 (0.15)	66	17 ± 5
2–6 mo	66	24 ± 4 ^a^	72	24 ± 5 ^a^	68	24 ± 5 ^a^	0.72 (0.70)	66	20 ± 5

* SF, standard formula; α-lac-EW, experimental formula with α-lactalbumin-enriched whey; CGMP-RW, experimental formula with reduced CGMP whey. ^1^ FF groups compared by one-way ANOVA, post hoc Bonferroni. ^(2)^ FF groups compared by one-way ANCOVA, post hoc Bonferroni, adjusted for weight gain during pregnancy, gestational diabetes, maternal smoking during pregnancy, maternal and paternal BMI. ^3^ Mean ± SD (all such values). ^a^ Significantly different vs. BF (*p* < 0.05).

**Table 6 nutrients-15-01010-t006:** Energy efficiency in infants fed formula (SF, α-lac-EW or CGMP-RW) * during the intervention, ITT population.

		SF		α-lac-EW		CGMP-RW	*p*-Value ^1^
	*n*		*n*		*n*		
Weight gain, g/100 kcal							
2–4 mo	69	4.23 ± 1.06 ^2^	74	4.47 ± 0.85	71	4.40 ± 0.78	0.28
4–6 mo	66	3.14 ± 0.95	67	2.96 ± 0.84	67	3.24 ± 0.87	0.19
2–6 mo	64	3.72 ± 0.82	67	3.76 ± 0.71	67	3.88 ± 0.57	0.39
Weight gain, g/g protein							
2–4 mo	69	1.92 ± 0.48 ^a,b^	74	2.56 ± 0.49	71	2.48 ± 0.44	<0.0001
4–6 mo	66	1.43 ± 0.43 ^a,b^	67	1.70 ± 0.49	67	1.83 ± 0.5	<0.001
2–6 mo	64	1.69 ± 0.38 ^a,b^	67	2.16 ± 0.41	67	2.19 ± 0.33	<0.0001
Length gain, mm/100 kcal							
2–4 mo	69	0.15 ± 0.03	74	0.16 ± 0.03	71	0.15 ± 0.03	0.094
4–6 mo	66	0.11 ± 0.04	67	0.11 ± 0.03	67	0.11 ± 0.03	0.93
2–6 mo	64	0.13 ± 0.02	67	0.14 ± 0.02	67	0.13 ± 0.02	0.41
Length gain, mm/g protein							
2–4 mo	69	0.068 ± 0.01 ^a,b^	74	0.092 ± 0.02	71	0.082 ± 0.02	< 0.001
4–6 mo	66	0.048 ± 0.017 ^a,b^	67	0.062 ± 0.02	67	0.061 ± 0.02	<0.001
2–6 mo	64	0.059 ± 0.01 ^a,b^	67	0.078 ± 0.01	67	0.075 ± 0.01	<0.001

* SF, standard formula; α-lac-EW, experimental formula with α-lactalbumin-enriched whey; CGMP-RW, experimental formula with reduced CGMP whey. ^1^ FF groups compared by one-way ANOVA, post hoc Bonferroni. ^2^ Mean ± SD (all such values). ^a^ SF vs. α-lac-EW. ^b^ SF vs. CGMP-RW.

**Table 7 nutrients-15-01010-t007:** Mean concentration (unadjusted) of blood urea nitrogen (BUN), s-IGF-1, insulin, c-peptide, leptin, leptin receptor, free leptin receptor index (FLI), hemoglobin (Hb), ferritin and iron in infants fed formula (SF, α-lac-EW or CGMP-RW) * or breast milk (BF) during the intervention, ITT population, and for BUN in the randomized subgroup.

		SF		α-lac-EW		CGMP-RW	*p*-Value ^1 (2)^		BF
	*n*		*n*		*n*			*n*	
BUN (mg/dL) ^3^									
2 mo	50	13.8 ± 4.2 ^4,a^	50	14.2 ± 5.0 ^a^	50	13.5 ± 4.1	0.72	50	11.4 ± 4.1
4 mo	50	15.1 ± 4.7 ^a,b,c^	50	11.0 ± 4.0	50	11.1 ± 3.8	<0.0001	50	9.8 ± 4.5
6 mo	50	14.5 ± 5.0 ^a,b,c^	50	11.6 ± 3.9	50	12.2 ± 4.7	0.004	50	10.3 ± 3.8
IGF-1 (µg/L)									
2 mo	80	87.2 ± 19.7	74	93.5 ± 18.2 ^a^	74	88.7 ± 16.8	0.094 (0.16)	76	84.2 ± 20.0
4 mo	67	71.8 ± 22.7 ^a^	72	71.4 ± 20.6 ^a^	67	69.1 ± 22.1 ^a^	0.74 (0.73)	69	55.0 ± 18.7
6 mo	67	60.2 ± 20.3 ^a^	69	62.9 ± 23.0 ^a^	65	61.1 ± 18.6 ^a^	0.74 (0.92)	68	44.8 ± 15.2
Insulin (mIU/L)									
2 mo	80	11.8 ± 7.8 ^a^	74	9.9 ± 5.2	74	11.5 ± 6.9 ^a^	0.19 (0.42)	75	8.4 ± 5.4
4 mo	67	8.7 ± 5.2 ^a^	72	8.1 ± 4.6 ^a^	67	8.7 ± 6.0 ^a^	0.72 (0.68)	68	4.9 ± 2.5
6 mo	67	6.8 ± 4.4 ^a^	69	6.7 ± 5.2 ^a^	65	7.1 ± 4.7 ^a^	0.85 (0.83)	68	4.0 ± 2.7
C-peptide (nmol/L)									
2 mo	80	0.69 ± 0.23 ^a^	74	0.66 ± 0.20 ^a^	74	0.72 ± 0.22 ^a^	0.26 (0.13)	74	0.53 ± 0.20
4 mo	67	0.63 ± 0.22 ^a^	72	0.61 ± 0.22 ^a^	67	0.63 ± 0.24 ^a^	0.90 (0.84)	68	0.43 ± 0.16
6 mo	67	0.53 ± 0.19 ^a^	69	0.54 ± 0.25 ^a^	65	0.55 ± 0.22 ^a^	0.95 (0.97)	68	0.37 ± 0.17
Leptin (ng/mL)									
2 mo	68	6.9 ± 3.6	63	6.4 ± 2.9	70	6.6 ± 3.9	0.74 (0.54)	62	7.3 ± 5.2
4 mo	62	5.3 ± 2.8	60	6.1 ± 3.7	62	6.3 ± 3.4	0.22 (0.079)	59	6.5 ± 4.3
6 mo	61	4.9 ± 2.6	64	5.3 ± 3.7	60	5.2 ± 2.7	0.71 (0.57)	58	4.7 ± 2.6
Soluble leptin receptor (ng/mL)									
2 mo	73	28.1 ± 4.8 ^a^	64	27.7 ± 6.0 ^a^	71	28.3 ± 5.6 ^a^	0.83 (0.74)	67	32.0 ± 7.1
4 mo	64	42.5 ± 9.1	63	39.4 ± 7.9	62	39.8 ± 9.5	0.081 (0.26)	62	42.2 ± 9.6
6 mo	62	46.5 ± 9.1	66	42.9 ± 11.0	61	43.4 ± 9.3	0.081 (0.16)	60	45.2 ± 9.4
Free leptin index ^5^									
2 mo	68	0.25 ± 0.14	63	0.24 ± 0.12	70	0.25 ± 0.18	0.86 (0.49)	62	0.26 ± 0.2
4 mo	62	0.13 ± 0.09 ^c^	60	0.16 ± 0.11	62	0.18 ± 0.1	0.054 (0.015)	59	0.17 ± 0.15
6 mo	61	0.11 ± 0.08	64	0.14 ± 0.12	60	0.13 ± 0.07	0.38 (0.27)	58	0.12 ± 0.11
Hb (g/L)									
2 mo	68	111.5 ± 13.0 ^a^	59	115.2 ± 15.2	67	113.5 ± 13.2	0.33	66	119.4 ± 12.0
4 mo	58	115.5 ± 8.3	61	115.8 ± 8.2	54	115.8 ± 8.4	0.97	61	116.5 ± 9.1
6 mo	56	116.4 ± 7.0	65	117.4 ± 8.3	59	115.3 ± 7.5	0.34	60	115.8 ± 7.6
Ferritin (µg/L)									
2 mo	78	309.5 ±126.8 ^a^	71	310.2 ± 144.9 ^a^	74	353.2 ± 157.3	0.11	71	396.8 ± 154.7
4 mo	66	106.8 ± 61.4 ^a^	68	122.0 ± 65.5 ^a^	65	145.6 ± 93.9 ^c^	0.013	66	172.5 ± 80.8
6 mo	65	74.6 ± 41.2	69	90.4 ± 60.6	65	91.1 ± 48.2	0.11	65	75.4 ± 45.3
Iron (µmol/L)									
2 mo	78	17.1 ± 4.7	71	17.8 ± 4.3	74	17.4 ± 4.2	0.65	71	16.1 ± 3.9
4 mo	66	10.0 ± 2.9	68	11.0 ± 3.7 ^a^	65	10.2 ± 3.1	0.12	66	9.2 ± 2.2
6 mo	65	9.1 ± 3.5	69	9.5 ± 3.3	65	9.7 ± 2.9 ^a^	0.53	65	8.2 ± 2.2

* SF, standard formula; α-lac-EW, experimental formula with α-lactalbumin-enriched whey; CGMP-RW, experimental formula with reduced CGMP whey. ^1^ Formula groups compared by one-way ANOVA, post hoc Bonferroni, *p*-value from this analysis. ^(2)^ Formula groups compared by one-way ANCOVA, post hoc Bonferroni, adjusted for weight gain during pregnancy, gestational diabetes, maternal smoking during pregnancy, maternal and paternal BMI, *p*-value from this analysis in parentheses. ^3^ Analyzed in subpopulation. ^4^ Mean ± SD. ^5^ Leptin/soluble leptin receptor. ^a^ Significantly different vs. BF (*p* <0.05). ^b^ SF vs. α-lac-EW. ^c^ SF vs. CGMP-RW.

## Data Availability

The data from this study are available on request from the corresponding author.

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
