# Peer review of "Low-Protein Formulas with Alpha-Lactalbumin-Enriched or Glycomacropeptide-Reduced Whey: Effects on Growth, Nutrient Intake and Protein Metabolism during Early Infancy: A Randomized, Double-Blinded Controlled Trial"

_nutrients, 2023, doi:10.3390/nu15041010_

Round 1

Reviewer 1 Report

Thank you for the opportunity to review this manuscript which examines multiple growth and biochemical measurements of infants in a randomized trial of 2 protein-modified infant formulas compared with standard formula and a breastfeeding reference group. The comparison of these two formulas with enhanced protein quality, yet reduced quantity, is novel and an important step towards improving infant formula and the health of the infants that consume them. This was a well-designed and rigorous study that assessed numerous relevant clinical and biochemical factors. The methods were appropriate, thorough, and clearly described. The outcomes measured were comprehensive, providing a complete picture of differences resulting from receipt of the experimental formulas compared to the standard formula and BF infants. Likewise, the discussion examined each detail of the authors’ findings along with an appropriate comparison/contrast with findings in existing literature. I believe this manuscript would be of great interest to readers of Nutrients, and an important contribution to the literature. I have a few minor comments.

General comments:

1.     What was the justification for selecting the protein concentration for the experimental formulas? I wondered why the protein was not lower than 1.75g/100kcal given that other studies have shown that even lower concentrations of formula (1.6g/100 kcal or below) may be safe. Perhaps the age of the infants at enrollment, and some other factors may have been considerations in determining this? Some comment about this would be appreciated.

2.     Any comments/discussion about why recruitment took so much longer than expected?

Minor comments, primarily related to grammar/formatting:

1.     Abstract: It would help to know how many were allocated to each of the 3 groups, as well as the size of the BF reference group in the abstract.

2.     Figure 3, top left (4mo Essential AA): Here, Histidine is the only portion of the entire figure where the indicator of significance (the superscript a) is above the BF group. It should instead be above the other 3 groups as in the 6mo EAA portion of the figure.

3.     Discussion, Lines 456-457: This sentence could be worded more clearly, especially the second portion (and similar or higher than in BF infants).

4.     Discussion, Lines 523-524: The first half of this sentence is missing a verb.

Reviewer 2 Report

The study is of very high quality, carefully thought out and prepared in terms of methodology. The results are very detailed, all factors that may have a potential impact on the results of the study have been taken into account.

Alternatively, the readability of figure 3 could be improved by presenting the graphs in one column instead of two, but the change is not essential.
